# Evolution and structure of clinically relevant gene fusions in multiple myeloma

Steven M. Foltz [1,2], Qingsong Gao [1,2], Christopher J. Yoon[1,2], Hua Sun[1,2], Lijun Yao[1,2], Yize Li[1,2], Reyka G. Jayasinghe[1,2], Song Cao[1,2], Justin King[1], Daniel R. Kohnen[1], Mark A. Fiala[1], Li Ding[1,2,3,4✉] & Ravi Vij[1,4✉]

Multiple myeloma is a plasma cell blood cancer with frequent chromosomal translocations leading to gene fusions. To determine the clinical relevance of fusion events, we detect gene fusions from a cohort of 742 patients from the Multiple Myeloma Research Foundation CoMMpass Study. Patients with multiple clinic visits enable us to track tumor and fusion evolution, and cases with matching peripheral blood and bone marrow samples allow us to evaluate the concordance of fusion calls in patients with high tumor burden. We examine the joint upregulation of *WHSC1* and *FGFR3* in samples with t(4;14)-related fusions, and we illustrate a method for detecting fusions from single cell RNA-seq. We report fusions at *MYC* and a neighboring gene, *PVT1*, which are related to *MYC* translocations and associated with divergent progression-free survival patterns. Finally, we find that 4% of patients may be eligible for targeted fusion therapies, including three with an *NTRK1* fusion.

[1] Department of Medicine, Washington University in St. Louis, St. Louis, MO 63110, USA. [2] McDonnell Genome Institute, Washington University in St. Louis, St. Louis, MO 63108, USA. [3] Department of Genetics, Washington University in St. Louis, St. Louis, MO 63110, USA. [4] Siteman Cancer Center, Washington University in St. Louis, St. Louis, MO 63110, USA. ✉email: lding@wustl.edu; rvij@wustl.edu

Fusions are a type of somatic alteration leading to cancer associated with up to 20% of cancer morbidity[1,2]. Translocations, copy number changes, and inversions can lead to fusions, dysregulated gene expression, and novel molecular functions. Fusions occur and have oncogenic roles in hematological, soft tissue, and solid tumors. Fusion rates differ across cancer types, and fusions may define some cancer types, such as BCR--ABL1 in chronic myeloid leukemia. A balanced translocation t(9;22) leads to BCR--ABL1, producing a hybrid protein with constitutive ABL1 kinase domain activation, signaling cell division, and avoiding apoptosis. Imatinib inhibits the BCR--ABL1 protein hybrid and in 2001 became the first FDA-approved drug to specifically target a fusion protein[2].

Multiple myeloma (MM) is the second most common blood cancer (10% of blood cancers, 1–2% of all cancers) and involves the clonal proliferation of bone marrow (BM) plasma cells, which are fully differentiated B cells. B cells produce a diverse repertoire of antibodies through genomic alterations at immunoglobulin (Ig) loci, including VDJ recombination, somatic hypermutation, and class switch recombination. Aberrant class switch recombination may result in translocations upregulating oncogenes. Ig enhancers get repurposed to drive oncogene expression, myeloma tumorigenesis, and clonal expansion[3].

Tumor initiating genomic changes may already be present at the pre-malignant stages of MM include monoclonal gammopathy of undetermined significance and smouldering MM. Primary genomic events in MM distinguish patient groups having hyperdiploidy (HRD, ~50%) and non-hyperdiploidy (non-HRD). Non-HRD patients typically have a different primary event, like an Ig translocation. CCND1 (chr11) and WHSC1 (chr4) are the two most common translocation partners of IGH (chr14). Patients may have both HRD and translocation events, and secondary events like t(8;14) dysregulating MYC are associated with progression[4,5].

Previous studies used RNA-seq to catalog fusion events from over 9000 patients and 33 cancer types from The Cancer Genome Atlas (TCGA)[6–8]. False positives due to library preparation or bioinformatic errors must be filtered. Overlapping fusion calls from multiple tools can establish concordance. Low expression or low quality RNA may cause false negatives, and translocations may affect expression but not produce detectable fusion transcripts. In myeloma, plasma cell Ig gene expression dominates the transcriptome and masks lower expression fusions. Multi-omic approaches with DNA and RNA resolves some limitations[2].

Large-scale sequencing efforts to understand multiple myeloma have demonstrated genomic heterogeneity beyond primary copy number and translocation events[9–12]. Several fusion detection studies show complementary results. Cleynen et al. detected gene fusions from 255 newly diagnosed MM patients, finding significant relationships between fusions and gene expression, hyperdiploidy, and survival, and identifying recurrent fusion gene partners[13]. Nasser et al. analyzed MMRF CoMMpass RNA-seq data, reconstructed Tophat-Fusion transcripts, and validated fusions with WGS[14]. Lin et al. used targeted RNA-seq in 21 MM patients, finding several novel fusions with disease relevance[15]. Morgan et al. used targeted sequencing of kinases to understand how translocations dysregulate kinase activity in MM[16].

Here, we extend previous efforts by focusing on the clinical implications and evolution of fusions across multiple time points. We leverage RNA and DNA sequencing as well as clinical data types to analyzed fusion genes we detected from the Multiple Myeloma Research Foundation (MMRF) CoMMpass Study. We analyze fusion genes and gene expression patterns from 742 multiple myeloma patients (806 samples). Patient samples from serial clinic visits enable tumor evolution profiles using fusions and mutations. Further, from patients with both BM and peripheral blood samples collected at the same time, we quantify the concordance of their fusion profiles. We demonstrate fusion event detection at single cell resolution using barcoded scRNA-seq data, pointing to future development of fusion methods. We explore the prognostic relevance of fusions by analyzing progression-free survival and find that those with IGH--WHSC1 or PVT1--IGL fusions have significantly worse outcomes. 4% of patients have a fusion annotated as a drug target in a public database.

## Results

**Fusion calling pipeline and clinical characteristics**. We detected gene fusions from 742 patients from the MMRF CoMMpass Study (see Data availability), combining RNA and DNA sequencing data with clinical information to form a landscape of fusion events (Fig. 1, Supplementary Fig. 1, Supplementary Data 1–3). We ran five fusion detection tools, implemented strict filtering criteria, and quantified gene expression to correlate with gene fusions (see "Methods"). We used WGS to detect structural variants and copy number changes potentially related to fusions. Sequencing-based FISH (seq-FISH) results showed major translocations and copy number changes such as hyperdiploidy[17]. We defined a primary sample for each patient as the earliest available sample and favored BM over peripheral blood (PB) (740 BM, 2 PB). For 97.2% of patients (721/742 patients), the primary sample corresponded with the pre-treatment clinic visit. 53 patients had RNA-seq from multiple samples (BM and PB from the same visit or data from serial visits), for a total of 806 RNA-seq samples. Results come from primary samples only, unless otherwise stated.

The cohort ranged from 27 to 93 years old (median 63) (Supplementary Data 1). Patients were spread evenly across ISS Stage, with 34.7% of patients from Stage I (247/711 patients with annotated stage), 35.7% Stage II (254/711), and 29.5% Stage III (210/711). Follow-up for progression-free survival ranged from 8 days to 5.7 years (median 2.23 years) with 60.3% of patients progressing (402/667 patients with PFS). Follow-up for overall survival ranged from 8 days to 6.43 years (median 3.19 years) with 27.4% of patients dying (182/665 patients with OS). ISS Stages I, II, and III patients had median PFS of 3.85 years, 2.47 years, and 1.76 years, respectively. 58.1% of patients showed a hyperdiploidy (373/642 patients with annotated HRD status). 77.1% of patients had ancestry reported as White (512/664 patients with annotated ancestry), 15.8% Black (105/664), and 7.1% Other (47/664). Most patients were treated initially with a proteasome inhibitor (bortezomib or carfilzomib) and an immunomodulatory drug (IMID) (68.4%). Others received a proteasome inhibitor-based regimen (25.9%) or an IMID (5.7%). 41.4% of patients received a BM transplant (305/737 with transplant annotated) during first-line therapy. Supplementary Data 1 summarizes clinical information.

**Immunoglobulin gene fusions are most frequent**. IGH--WHSC1 was the most common fusion reported; it results from t (4;14) typically observed in 15% of patients[4]. IGH--WHSC1 or WHSC1--IGH were found in 12.4% of samples (92/742 samples). 79.7% of IGH--WHSC1 fusions showed WGS support (47/59 patients with WGS data) (see "Methods"). Fig. 1b shows the top recurrent fusions with at least one fusion supported by WGS. Ig fusions (IGH, IGK, or IGL) were reported frequently (35.6%, 1102/3094 fusions) with upregulated partner genes.

Our pipeline reported fusions between Ig loci and MYC or its downstream neighbor PVT1. MYC or PVT1 was usually the 5′ gene and paired with IGH, IGK, or IGL, including 18 samples with MYC--IGL, 11 with PVT1--IGL, 6 with PVT1--IGH, and 3 with PVT1--IGK (Fig. 1b). One sample had IGH--MYC and one had IGL--PVT1. Past reports show MYC translocations with all three

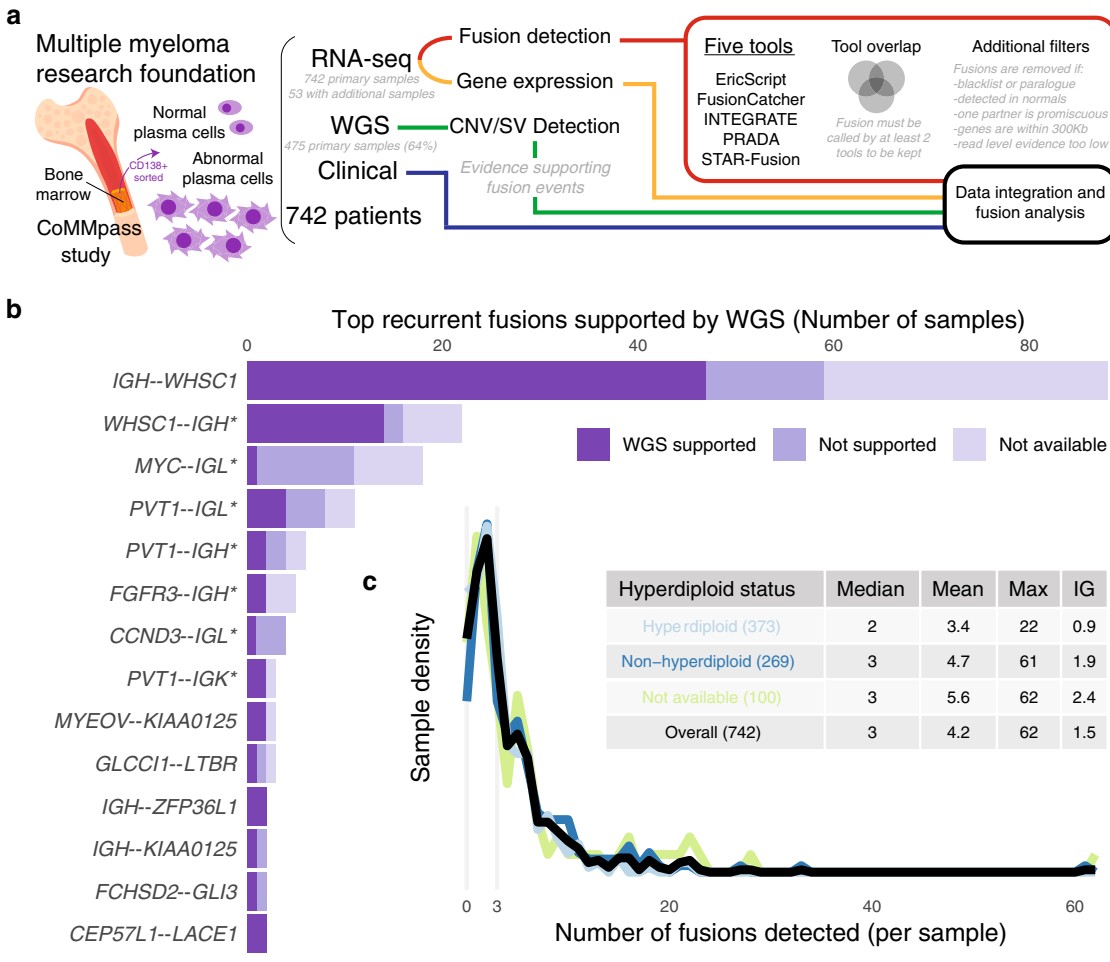

**Fig. 1 Overview of pipeline and fusions reported. a** Project pipeline. **b** Recurrent fusions with at least one sample's fusion supported by WGS. An asterisk (*) annotation refers to reciprocal fusions with the opposite orientation of the canonical fusion led by an Ig partner gene. **c** Number of fusions detected per sample, stratified by hyperdiploid status. Source data and scripts are available at https://doi.org/10.6084/m9.figshare.11941494.

Ig loci[18]. However, previous multiple myeloma fusion studies hypothesized that *MYC* fusions with Ig would not be detected from RNA-seq if there were no hybrid transcript generated after the translocation[13]. Further study is necessary to determine whether these reported fusions are true events, biological by-products, or bioinformatic artifacts, and whether they confer functional or clinical significance. This will complement recent work showing the dysregulation of both *MYC* and *PVT1* in the presence of super-enhancer translocations[19,20].

The number of fusions reported per sample ranged from 0 to 62 (median 3) (Fig. 1c, Supplementary Fig. 1a), similar to breast, glioblastoma, ovarian, and prostate cancers from TCGA[8]. Hyperdiploid samples had significantly fewer fusions reported than non-HRD samples (HRD mean 3.4, non-HRD 4.7, Mann–Whitney U test two-sided $p$ value $6.71 \times 10^{-3}$). There were also significantly fewer Ig fusions between those groups (HRD mean 0.9, non-HRD 1.9, Mann–Whitney U test two-sided $p$ value $7.88 \times 10^{-8}$). We required two or more tools to agree upon a particular fusion call. We removed 18 highly recurrent IGL fusions with low WGS support (see "Methods"). After filtering, the overall WGS support rate was 22.3% (comparable to a previously reported pan-cancer support rate of 32.5% from samples with similar WGS coverage)[8]. Most fusions were called by two tools (73.3%, 2269/3094), while 17.9% (555/3094) were called by three or four tools, and 8.7% were called by all five tools (270/3094) (Supplementary Fig. 1b).

**Fusion gene expression highlights multiple myeloma biology.** Fusions may be associated with expression changes of the partner genes. We defined a sample's expression percentile for each gene as their expression level relative to primary samples at that gene (see "Methods"). The median fusion expression percentile of a gene is the median expression percentile of samples with a fusion involving that gene. We identified 51 genes significantly over-expressed in fusion samples (FDR < 0.05 or median fusion expression percentile >0.9) (Supplementary Data 4). Of those, nine are cancer-related genes from any cancer type annotated as a driver, drug target, kinase, oncogene, or tumor suppressor (Fig. 2a), including *FGFR3* (12 samples), *MAPKAPK2* (5), *MYC* (19), *NTRK1* (3), *PAX5* (3), *PIM3* (3), *RARA* (3), *TXNIP* (7), and *WHSC1* (97)[21]. Expression levels may also identify samples with a false negative fusion call. 12 samples have outlier *WHSC1* over-expression but no *WHSC1* fusion reported, representing false negative *IGH--WHSC1* fusions or indicating samples with t(4;14) but no fusion product formed. Of those 12 samples, 50% (5/10 with seq-FISH) have a *WHSC1* translocation with expression percentile over 0.87. The tumor etiology of samples with high gene expression but no fusion calls may still involve upregulated gene activity. Since gene expression is itself relevant to cancer biology and drug targeting, fusion analysis should always be paired with gene expression.

Samples with fusions involving kinases, oncogenes, and tumor suppressors show different trends in expression levels of those

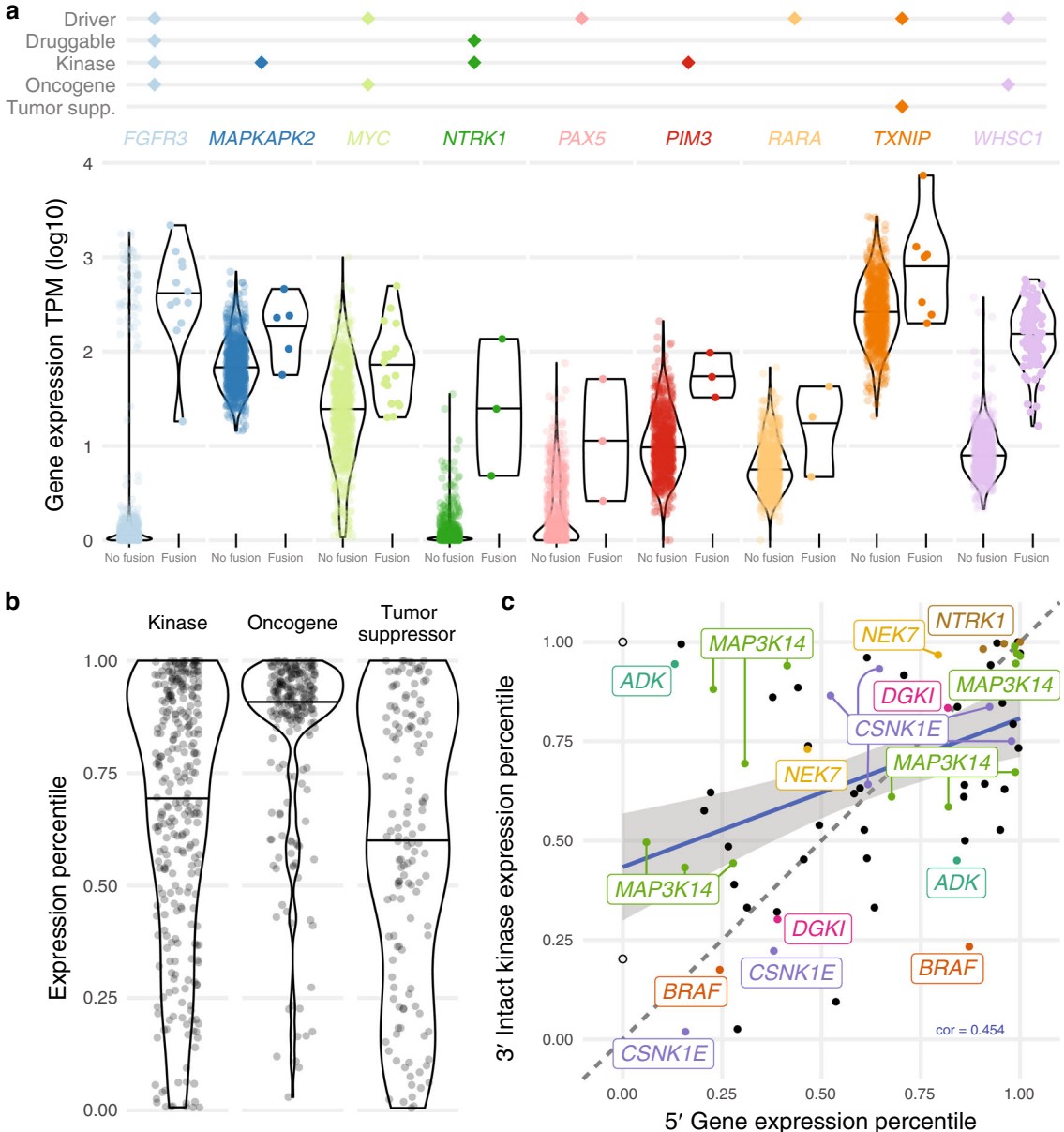

**Fig. 2 Expression of cancer-related genes. a** Significantly overexpressed fusion genes. Violin plots defined as: *FGFR3* no fusion (n = 730, minimum = 0.00, maximum = 3.26, median = 0.05), fusion (12, 1.26, 3.34, 2.58); *MAPKAPK2* no fusion (737, 1.16, 2.85, 1.83), fusion (5, 1.75, 2.66, 2.36); *MYC* no fusion (723, 0.03, 3.00, 1.39), fusion (19, 1.30, 2.69, 1.89); *NTRK1* no fusion (739, 0.00, 1.55, 0.03), fusion (3, 0.68, 2.13, 1.39); *PAX5* no fusion (739, 0.00, 1.88, 0.11), fusion (3, 0.42, 1.71, 1.05); *PIM3* no fusion (739, 0.00, 2.32, 0.99), fusion (3, 1.51, 1.99, 1.73); *RARA* no fusion (739, 0.00, 1.83, 0.75), fusion (3, 0.67, 1.63, 1.31); *TXNIP* no fusion (735, 1.31, 3.43, 2.41), fusion (7, 2.3, 3.87, 3); *WHSC1* no fusion (645, 0.33, 2.57, 0.91), fusion (97, 1.21, 2.77, 2.19). All data points from 0 to 100th percentile are shown. **b** Expression percentile distribution for different gene classes. Violin plots defined as: Kinase (n = 298, minimum = 0.01, maximum = 1.00, median = 0.74); Oncogene (224, 0.03, 1.00, 0.92); Tumor Suppressor (137, 0.01, 1.00, 0.61). All data points from 0 to 100th percentile are shown. **c** 5′ and 3′ gene expression of fusions with intact 3′ kinase genes. The two empty circles have no expression value for the 5′ gene (IGH, IGL). Labels refer to genes appearing more than once. Source data and scripts are available at https://doi.org/10.6084/m9.figshare.11941494.

genes (Fig. 2b). Gene expression of fusion oncogenes tended to be higher relative to other samples, fitting the biological context of oncogenes being deleterious when upregulated. Tumor suppressors, which may be disrupted in cancer in many different ways, displayed no trend of up- or downregulation. Kinases showed a skewed preference toward upregulation and are an important type of gene with implications for cancer development and drug targeting. We investigated the correlation between 5′ and 3′ partner gene expression when the 3′ partner gene is a kinase and contains an intact kinase domain (see "Methods") (Fig. 2c, Supplementary Data 5). In this subset of fusion partners, the

positive correlation between 5′ and 3′ gene expression is somewhat higher than that of the overall background (0.454 vs. 0.352), indicating a pattern of selection for overexpressed kinase fusion partners. Recurrent 3′ kinases with intact domains included: *MAP3K14* (13 patients), *CSNK1E* (7), *NTRK1* (3), *ADK* (2), *BRAF* (2), *DGK1* (2), and *NEK7* (2).

We tested for associations between clinical data (including age, sex, ancestry, ECOG performance, ISS stage, bone lesions, plasmacytoma, BM plasma cell percentage, and LDH) and fusion genes observed in three or more samples (see "Methods"). After FDR correction and assessment of model fit, no clinical measures

were significantly associated with fusion events. To understand the relationship between fusion events and prognosis in this cohort, we analyzed survival in patients with and without particular fusions or fusion genes. We created baseline PFS multivariate Cox proportional hazards models, including disease stage and patient age as covariates. For each fusion or fusion gene observed in ten or more samples, we added the event as a covariate and tested for significant improvement in model fit using a chi-squared test. WHSC1 and PVT1 fusions were significantly associated with worse prognosis (Supplementary Fig. 2a–c). The estimated hazard ratio (HR) for a WHSC1 fusion was 1.43 (95% CI 1.07–1.90; two-sided z-score p value 0.0157). For PVT1 fusions, the estimated HR was 2.01 (95% CI 1.17–3.46; two-sided z-score p value 0.0114). For PVT1--IGL specifically, the HR estimate was 3.42 (95% CI 1.75–6.69; two-sided z-score p value 0.000324). After including R-ISS and common translocations as covariates in the model, no fusion events or fusion genes were significantly associated with PFS, likely due to confounding introduced by translocation events directly associated with fusions. Total fusion burden was associated with worse prognosis; each additional fusion was associated with a slight decrease in PFS (HR estimate 1.02; 95% CI 1.00–1.04; two-sided z-score p value 0.0178), after controlling for disease stage and patient age (Supplementary Fig. 2d).

Patients are stratified into risk groups by genomic events like amp(1q), del(17p), t(4;14), t(14;16), and t(14;20) using mSMART criteria[22]. Patients with multiple high-risk events have worse prognosis[23]. Walker et al. identified a subgroup of patients with especially poor outcomes having biallelic TP53 inactivation (for example, del(17p) and inactivating mutation) or Stage III disease and high copy number of CKS1B (1q21)[24]. In our data, we defined a double hit group of patients with both amp(1q) and del(17p). The median PFS time for this group was 581 days (19 patients, 14 progressed). Five patients with an additional t(4;14) event and IGH--WHSC1 fusion had median PFS of 142 days (5 patients, 4 progressed). Ongoing research with larger sample sizes and longer follow-up will enable more robust survival modeling utilizing genomic events to define progression and overall survival risk[25].

### Fusions from multiple time points highlight tumor evolution.
In total, 53 patients had additional samples allowing for within-patient comparisons across time (serial visits) or from different tissue sources (bone marrow, BM; peripheral blood, PB). In total, 45 patients had BM samples from serial visits, and we compared fusions from the first two visits (Fig. 3a, Supplementary Fig. 3). When initiating clonal fusion IGH--WHSC1 was detected at the earlier visit, it was always detected at the later visit (6/6 patients). In one patient (1/39 patients), IGH--WHSC1 was observed only at the later visit, but WHSC1 expression at the earlier visit was above the 98th percentile, indicating a likely t(4;14) and false negative fusion call.

For some samples with sufficient PB tumor burden, such as patients with plasma cell leukemia, both BM and PB samples had RNA-seq. In this subset, we compared fusions detected from both samples from the same visit (Fig. 3b) (10 patients, 11 visits). IGH--WHSC1 events were always detected in both or neither sample. Overall, more fusions were reported from BM samples than PB samples. We calculated the normalized Hamming distance between each pair of samples to quantify their overlap. Values ranged from 0.33 in pairs sharing 2 out of 3 fusions to 1 in completely discordant pairs. Previous studies have shown that tumor cells derived from peripheral blood have highly similar somatic mutation and copy number profiles[26]. Our comparison,

limited to a subset of patients with high tumor burden, quantifies the fusion landscape consistency between BM and PB samples.

Next, we considered the evolution of the fusion and mutation landscape between earlier and later clinic visits, especially in four patients illustrating different patterns of clonal changes (Fig. 3c–d). Analyzing the genetic changes and clonality structures that promote relapse remains important for understanding treatment response[27]. MMRF 1433 had many more fusions reported at Visit 2 compared with Visit 1 (Fig. 3c), and the appearance of ATM and other mutations at Visit 2 indicates a shift in clonal architecture (Fig. 3d). Low fusion expression at Visit 2 could indicate tumor heterogeneity or correspond to low tumor purity (66%). In MMRF 1496, the NRAS mutation at Visit 1 (VAF 0.673 with copy number loss) was not detected at Visit 3 (no mutation call or read-level evidence), meaning the NRAS mutant subclone was lost between visits. The CDC42BPB and MNAT1 fusions remained present, implying the hemizygous NRAS mutant subclone arose after or independently of those fusions. In MMRF 1656, there was one clonal missense mutation in kinase BCR and one important fusion event, TPM3--NTRK1. The absence of a known oncogenic driver mutation at Visit 1 may mean the NTRK1 fusion played a tumorigenic role and could have been an ideal drug target high on the tumor evolutionary tree. By Visit 4, mutations in FAM46C, FGFR3, and KRAS were detected at or above 50% VAF, indicating a strong clonal expansion of the new mutations after diagnosis. In contrast, another patient with an NTRK1 fusion, MMRF 2490, had clonal mutations in well-known myeloma tumor suppressors EGR1 and DIS3, meaning that targeting the NTRK1 fusion alone may not have been sufficient. Those mutations as well as expression levels of the fusion gene indicate tumor stability. Measures of fusion allele frequency useful for tracking clonal dynamics remain complicated by lower detection power and consistency compared with mutations; confident assessment of fusion VAF from expression data is an area of ongoing research and may benefit from cross-platform data integration. Further, the clonal resolution possible from bulk RNA-seq can be improved by methods that detect fusion events from scRNA-seq data.

### Chimeric transcripts in scRNA-seq reveal single cell fusions.
Fusion detection from bulk RNA-seq returns a fusion list but little further resolution. To detect fusions in single cells or, more broadly, present in tumor subclones, we analyzed barcoded scRNA-seq data from in-house MM patients generated on the 10x Genomics Chromium platform 3′ scRNA-seq protocol. Previous MM studies utilized scRNA-seq to investigate variation in heterogenous tumors, and AML mutations in single cells illustrated tumor specificity and subclonality[28,29]. Our method detects chimeric transcripts associated with cell and molecule barcodes and map those to their cell of origin (see "Methods"). We analyzed scRNA-seq data from 5 MM patients (eight samples). Patients had known translocations that guided our discovery, including one t(4;14), one t(8;14), and three t(11;14). The results reflect trends learned from bulk analysis but with additional, informative detail (Supplementary Fig. 4). In samples with an initiating t(4;14), fusion events are readily detected and map to specific malignant plasma cell subclones. In the patient with a secondary t(8;14) event, the t(8;14) subclone appears to be lost at relapse, emphasizing patterns of tumor heterogeneity and treatment response. Finally, although evidence of t(11;14) events is often observed in RNA and scRNA-seq due to upregulation of CCND1, actual IGH--CCND1 fusion transcripts may not be present or reported at the RNA level, and we find a similar low detection rate of chimeric transcripts in scRNA.

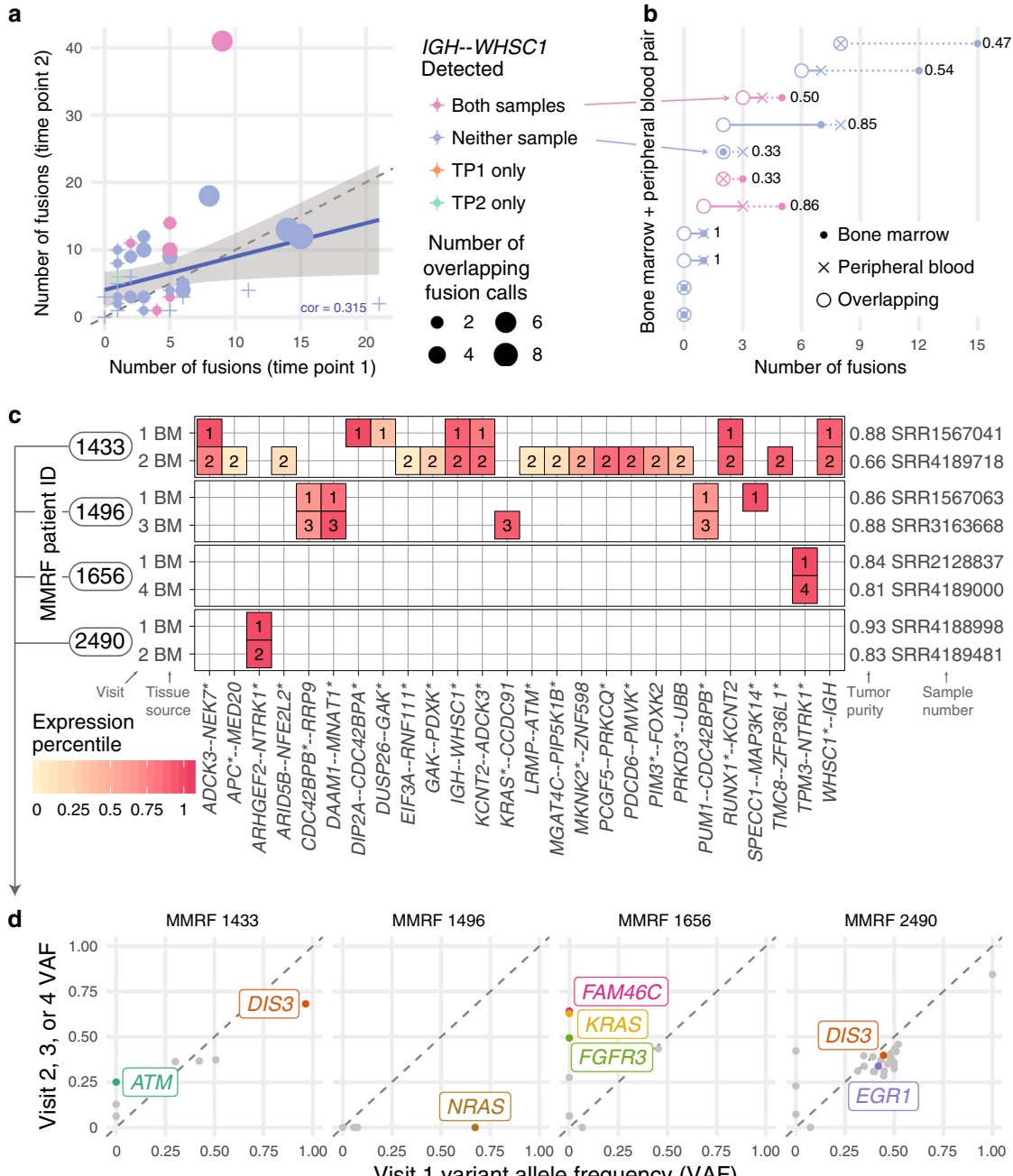

**Fig. 3 Fusions detected from multiple clinical samples and fusion evolution. a** Number of fusions called at serial clinic visits. Shaded region indicates a 95% confidence interval on the regression line. **b** Overlap of fusions called from bone marrow (BM) and peripheral blood (PB) from the same clinic visit with normalized Hamming distance (range 0–1, 0 = perfect overlap, 1 = completely discordant). **c** Fusions from cancer-related genes detected at serial clinic visits. **d** Somatic mutations from cancer-related genes detected at serial clinic visits. Genes frequently mutated in multiple myeloma are labeled. Source data and scripts are available at https://doi.org/10.6084/m9.figshare.11941494.

Quality control steps identified regions with high transcript overlap (see "Methods"). In these regions, true positive chimeric transcripts from real fusions may be detected in addition to chimeric transcripts attributed to high expression of certain genes. We confidently mapped one sample's *IGH--WHSC1* fusion events from non-overlapping genomic regions to single cells. This sample (Patient 27522, primary) comprised plasma cells (54.5%, 2477/4543 cells), monocytes (29.8%), B cells (6.6%), and CD4+ T cells, CD8+ T cells, and dendritic cells, each under 5% (Fig. 4a). We defined a high-confidence subpopulation of tumor cells harboring del(chr13) to evaluate the sensitivity of our approach. In that subpopulation, our non-overlap detection rate was 4.6%

(54/1166 tumor cells) (Fig. 4b). Furthermore, no fusions mapped to non-plasma cells. The expression pattern of *WHSC1* and *FGFR3* indicates upregulation across all plasma cells, although there is subregional variation (Supplementary Fig. 5a-b). Since t(4;14) and *IGH--WHSC1* are often clonal, our method showed overall low detection power, possibly reflecting the sparsity and positional bias of 3′ scRNA-seq sequencing or the stringency of our quality control.

Fusion-support reads from bulk and scRNA-seq reads mapped to similar exonic locations along the IGH region and *WHSC1* gene body (Fig. 4c) and illustrate some transcript heterogeneity. After t(4;14), transcription proceeds from chr14 (negative strand) (IGH

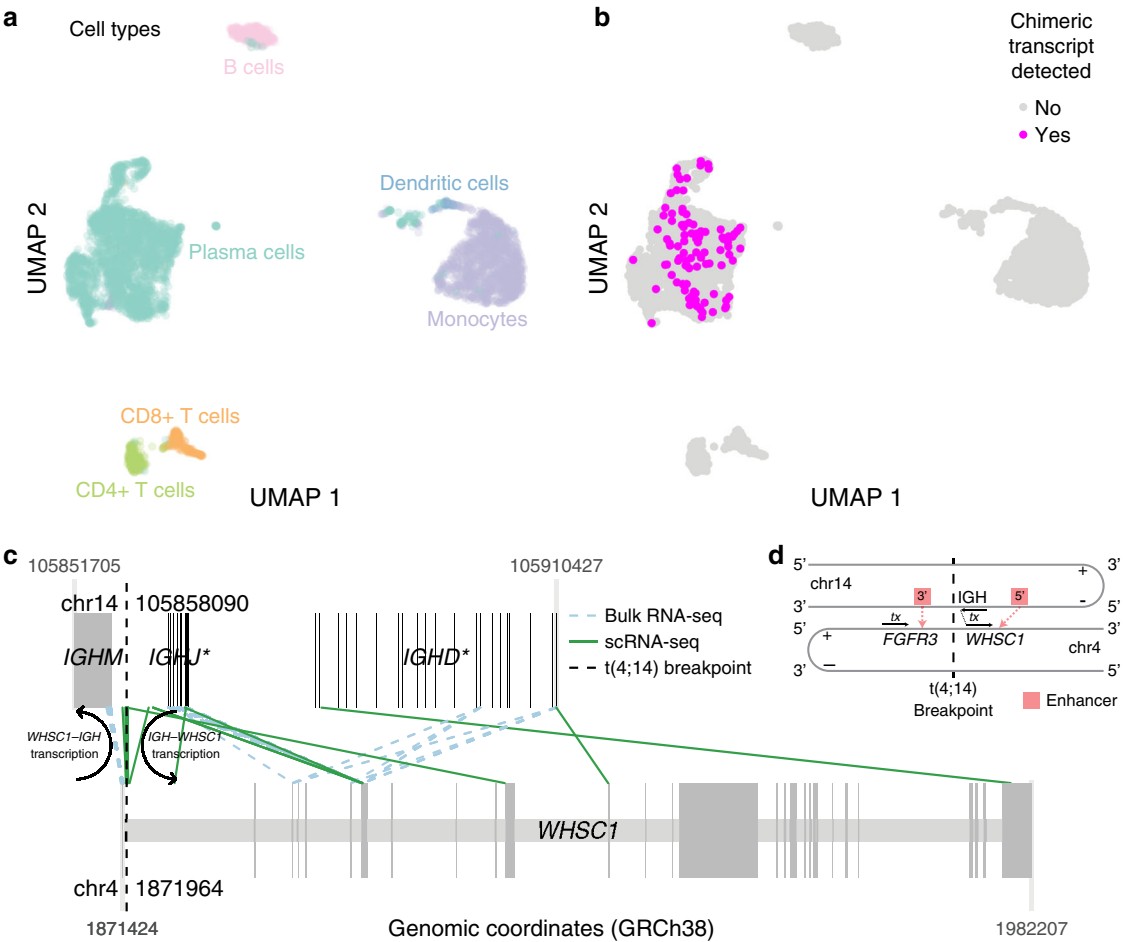

**Fig. 4 Single cell chimeric transcript detection. a** Cell types present from one patient's scRNA-seq sample (27522 primary disease stage, with t(4;14)). **b** Cells with chimeric transcripts detected from non-overlap regions. **c** Mapping location of paired-end (bulk) or same barcode (scRNA-seq) reads. Darker gray boxes indicate exons; lighter gray boxes span gene boundaries. **d** *IGH--WHSC1* fusion transcription model. Source data and scripts are available at https://doi.org/10.6084/m9.figshare.11941494.

region) to chr4 (positive strand) (*WHSC1*) (Fig. 4d). Reads mapping to the right of the t(4;14) breakpoint (vertical dotted black line) on both chromosomes (chr1) support *IGH--WHSC1*. Reads mapping to the left are transcribed in the opposite direction and support *WHSC1--IGH*. Reads from non-overlapping regions mapped to the *IGHM*, IGHJ, and IGHD regions of the IGH superlocus, precisely where *IGH--WHSC1* and t(4;14) were detected from bulk sequencing. (Supplementary Fig. 5c).

Despite the resolution gained from single end scRNA-seq, we lose the benefits of paired reads used for fusion detection from bulk data. Our method demonstrates the potential utility and feasibility of mapping fusions to individual cells. Long-term implications include better understanding of tumor heterogeneity, subclonality, and the relationship of fusion events with gene expression and somatic alterations. Continued methods development, both in sample sequencing and fusion detection, building upon this early work is necessary to improve single cell fusion mapping accuracy and sensitivity. Future methods and data, especially full-length transcript scRNA-seq data, will elucidate complex expression changes due to MM translocations and fusions, which have only been analyzed in bulk RNA-seq.

**IGH translocations lead to dysregulated *WHSC1* and *FGFR3*.** MM translocations juxtapose highly expressed immunoglobulin loci (IGH, IGK, and IGL) with oncogenes such as *WHSC1* and *MYC*, leading to upregulation and tumor selective advantage.

Neighboring genes may also be dysregulated through this process, like when *WHSC1* and *FGFR3* are both dysregulated with t(4;14). Typically, the t(4;14) translocation breakpoint on chr4 occurs between *WHSC1* and its upstream neighbor *FGFR3*. Previous studies showed that *WHSC1* and *FGFR3* are both upregulated in around 70% of patients while the remaining 30% only have high *WHSC1* expression[30]. In our data, 93 patients had a reported *IGH--WHSC1*, *IGH--FGFR3*, or reciprocal fusion; all had high *WHSC1* expression and 72.0% (67/93 patients) had *FGFR3* overexpression (Fig. 5a). No samples had *FGFR3* overexpression without t(4;14). Of patients with high *FGFR3* expression and mutation calls, 15.3% (9/59 patients) had somatic mutations in *FGFR3* (see "Methods"), all of which were copy number neutral at *FGFR3*. Interestingly, when we compared the DNA and RNA VAF of each *FGFR3* mutation, the RNA VAF was always 2–4 times higher than the DNA VAF, indicating a strong pattern of allele specific expression in all nine cases. We hypothesize that the *FGFR3* mutant allele expression is driven by the 3′ enhancer of IGH located on the same allele as the mutation. In this scenario, expression of the translocation allele dominates the expression landscape, and the RNA VAF reflects the proportion of translocation alleles with the *FGFR3* mutation.

We then used available WGS translocation breakpoint and CNV data available from 34 samples with a reported *IGH--WHSC1* fusion. We observed no relationship between *FGFR3* expression status and the location of genomic or fusion breakpoints (Supplementary

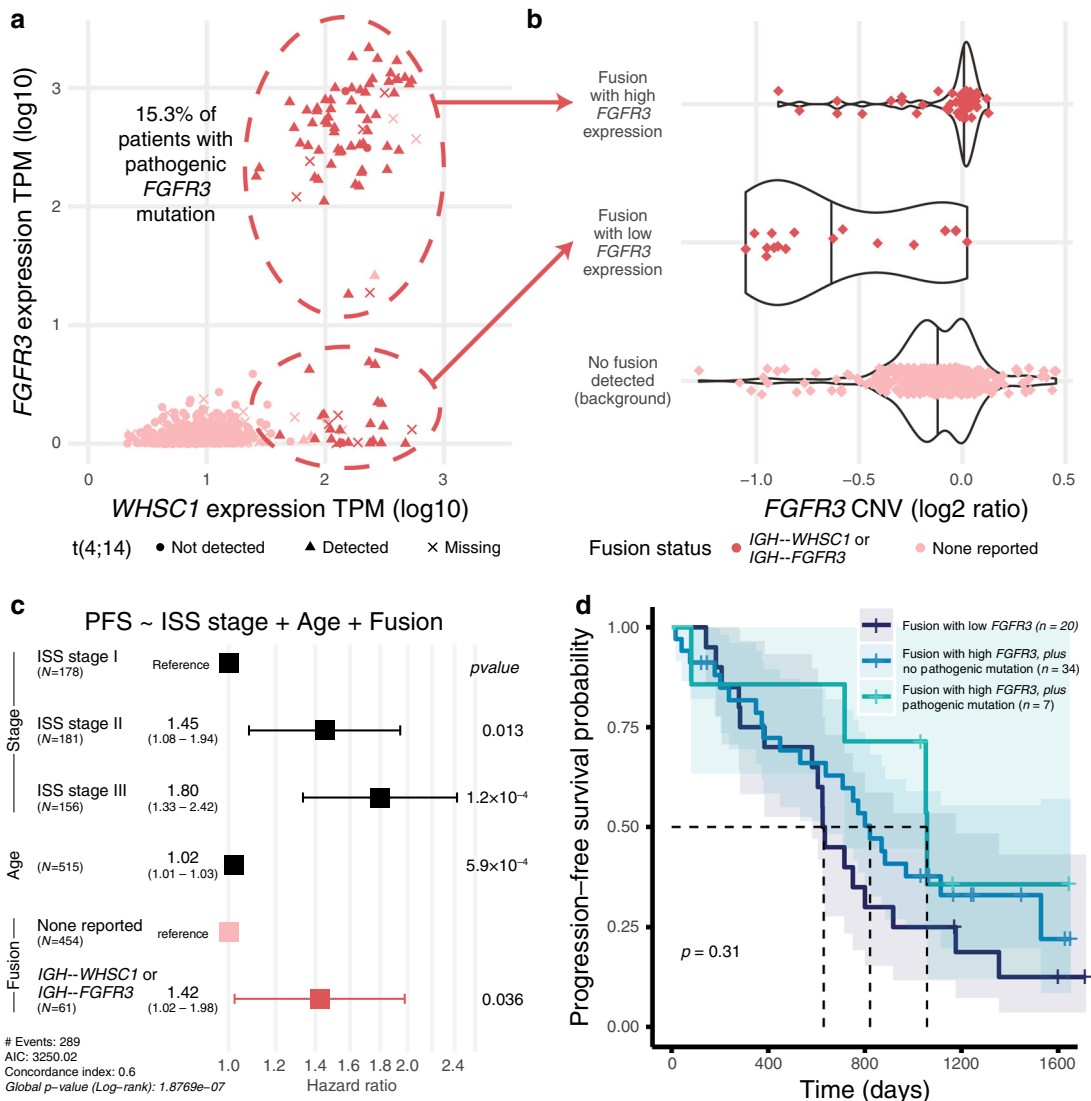

**Fig. 5 t(4;14) *WHSC1* and *FGFR3* expression and survival patterns. a** Coexpression of *WHSC1* and *FGFR3*, annotated with fusion and translocation status. **b** *FGFR3* copy number (log2 of the tumor/normal ratio). Violin plots defined as: Fusion with high *FGFR3* expression (n = 60, minimum = −0.89, maximum = 0.127, median = 0.013); Fusion with low *FGFR3* expression (18, −1.05, 0.024, −0.72); No fusion detected (background) (412, −1.28, 0.45, −0.13). All data points from 0 to 100th percentile are shown. **c**. Multivariate Cox proportional hazards progression-free survival model including disease stage, age, and fusion status. Error bars indicate a 95% confidence interval on each hazard ratio estimate. Covariate *p* values derived from z-scores are two-sided. **d** Kaplan–Meier curve stratified by *FGFR3* expression among fusion patients. Shaded regions indicate a 95% confidence interval on each survival curve. Significance *p* value was calculated by two-sided log-rank test and uncorrected for multiple comparisons. Source data and scripts are available at https://doi.org/10.6084/m9.figshare.11941494.

Fig. 6a). Fusion samples with low *FGFR3* expression had distinctly lower *FGFR3* copy number (Fig. 5b) while corresponding *WHSC1* copy number tended to remain neutral (Supplementary Fig. 6b), suggesting a loss of *FGFR3* after t(4;14) translocation[31]. Genomic breakpoints near IGH ranged over 0.27 Mb on chr14, while the chr4 genomic breakpoints ranged over 0.07 Mb, occurring both upstream of and within the gene body of *WHSC1*. As expected, *IGH--WHSC1* fusion breakpoints always occurred downstream of the genomic breakpoints on chr4, with three fusion breakpoint groups coalescing in the documented MB4-1, MB4-2, and MB4-3 regions of *WHSC1* (Supplementary Fig. 6c)[32].

Patients with pre-treatment *IGH--WHSC1* showed poorer PFS in a multivariate Cox proportional hazards model compared with patients with the same ISS stage and age (HR 1.42; HR 95% CI 1.02–1.98; two-sided z-score *p* value 0.035880) (Fig. 5c). Among patients with *IGH--WHSC1*, there was no difference in PFS

between those with high and low *FGFR3* expression (Fig. 5d, Supplementary Fig. 2e). For the few patients with pathogenic *FGFR3* mutation and available survival data (7 patients, 4 events), mutation status was not a significant model predictor, although the small sample size after stratification precludes any robust conclusion.

**MYC translocations lead to *MYC* and *PVT1* fusions.** Samples with *MYC* mutations or Ig fusions involving *MYC* or its downstream neighbor *PVT1* showed elevated *MYC* expression (Fig. 6a). Ten samples had a *MYC* mutation. *MYC* fusion breakpoints occurred across the *MYC* gene body while *PVT1* fusion breakpoints were located mostly at its 5′ end; Ig breakpoints ranged across each Ig region (Supplementary Fig. 7).

IGL translocations predict decreased survival in MM[18]. Kaplan–Meier curves for *PVT1--IGL* and *MYC--IGL* show that

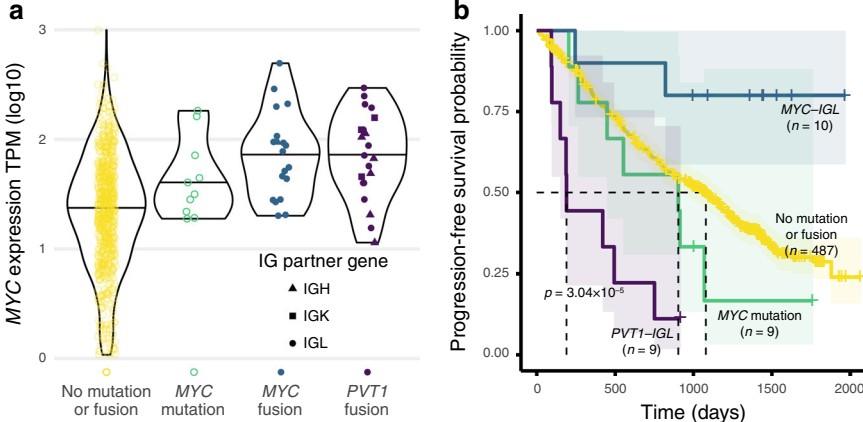

**Fig. 6 MYC translocation fusion partners and survival differences. a** *MYC* expression by *MYC* mutation or fusion status. Violin plots defined as: No mutation or fusion (n = 696, minimum = 0.03, maximum = 3.00, median =1.37); *MYC* mutation (10, 1.28, 2.26, 1.55); *MYC* fusion (19, 1.30, 2.69, 1.89); *PVT1* fusion (21, 1.06, 2.47, 1.85). All data points from 0 to 100th percentile are shown. **b** Kaplan–Meier curves stratified by *MYC* mutation or fusion status. Shaded regions indicate a 95% confidence interval on each survival curve. Significance *p* value was calculated by two-sided log-rank test and uncorrected for multiple comparisons. Source data and scripts are available at https://doi.org/10.6084/m9.figshare.11941494.

patients with *PVT1--IGL* had worse survival than the background (median PFS 190 days), while patients with *MYC--IGL* showed better survival with more censoring (median PFS not reached) (Fig. 6b). Further, only 18.2% of *PVT1--IGL* patients were ISS Stage I, while 43.8% of *MYC--IGL* patients were ISS Stage I. In a Cox model including ISS Stage and patient age, *PVT1--IGL* status had an estimated HR of 3.90 (95% CI 1.91–7.95; two-sided z-score *p* value 0.000181), while the *MYC--IGL* HR estimate was 0.26, (95% CI 0.06–1.05; two-sided z-score *p* value = 0.059018) (Supplementary Fig. 2f). Of the 15 patients with complete seq-FISH data and *MYC--IGL* or *PVT1--IGL*, 8 had *MYC--IGL* and 7 had *PVT1--IGL*. One of eight with *MYC--IGL* had t(8;22). Six of seven with *PVT1--IGL* had t(8;22). Thus, fusions annotated as *PVT1--IGL* may be more closely associated with t(8;22) than fusions annotated as *MYC--IGL*. *PVT1--IGL* has prognostic value to the extent that it is a proxy for t(8;22). Follow-up is needed to evaluate the source and relevance of these reported events. The *MYC/PVT1* relationship and its role in tumorigenesis remains an area of ongoing research.

*MYC* and *MYC* paralogs can be dysregulated through copy number amplification, viral integration, and translocation[33]. MM Ig translocations dysregulating MYC predict poor survival, and *MYC* can be downregulated by BET domain inhibitors[18,34]. One oncogenic role of lncRNA *PVT1* is to stabilize and upregulate MYC protein, promoting tumorigenesis[35]. In contrast, the *PVT1* promoter may compete with the *MYC* promoter, acting as a tumor suppressor[36]. *PVT1* promoter mutations may disrupt that *MYC* downregulation. Future studies will determine how genomic variation affects *MYC/PVT1* interactions. The *MYC* region is a hotbed of genomic rearrangement and instability. The underlying mechanisms contributing to the tumor evolutionary advantage of this complex pattern could be elucidated by ongoing and future studies, especially with haplotype-resolved copy number and translocation calls[37].

**Fusions are potential drug targets with prognostic relevance.** MM treatment often involves combination therapies, including alkylating agents, histone deacetylase inhibitors, immunomodulatory agents, monoclonal antibodies, proteasome inhibitors, and steroids[5]. Patients with actionable mutations in *BRAF*, *KRAS*, *NRAS*, *FGFR3*, or upregulation of *CCND1*, *CCND3*, and *MYC* may be eligible for targeted therapies[5].

We discovered 11 fusion genes reported in the Database of Evidence for Precision Oncology as potentially sensitive to drug treatment in other cancer types (Supplementary Fig. 8a)[38]. 4.0% of patients had a fusion annotated as druggable. We observed two patients with *BRAF* fusions, and *BRAF* fusions have shown some evidence of sensitivity to MEK pathway inhibitors in the absence of other drivers[39]. We found direct overlap of potentially druggable fusions in six cancer types (Supplementary Fig. 8b), pointing toward opportunities for tissue-agnostic clinical trials.

Kinase fusions are important across cancer types, especially since they may be sensitive to kinase inhibition. In our cohort, common kinase pathways with fusion genes included the NIK, MAPK, and RAS pathways. We compared intact 3′ kinase fusions from our cohort to those reported from a TCGA pan-cancer analysis (Supplementary Fig. 9) and found the same 3′ kinase fusions reported across 22 cancer types[8]. Fusions with *ADK*, *BRAF*, and *NTRK1* were reported repeatedly both in our cohort and in multiple cancer types.

*NTRK* genes, including *NTRK1*, encode cell surface neuro-trophin receptor tyrosine kinases. TRK fusions are a drug target in solid cancers, although TRK inhibition may lead to resistance mechanisms[40]. TRK fusions from hematological cancers were responsive to inhibition in cell culture and mouse modeling[41]. We found three patients with 3′ *NTRK1* fusions, each with an intact kinase domain (Supplementary Fig. 8c), and two had the same fusion detected at a later clinic visit (Supplementary Fig. 3). All three primary samples had strong WGS support for their fusion event. *NTRK1* fusion 5′ partners all came from the opposite strand of the same chr1, indicating that an inversion event may have brought the two genes together. There is also evidence of chr1 1q copy number amplification in these samples, highlighting overall genomic instability in the region. Each partner gene had expression in the 90th percentile or above, potentially driving *NTRK1* activity higher (Fig. 2c), and *NTRK1* was overexpressed in each case (Fig. 2a), leading to upregulation of downstream pathways.

APOBEC signature is associated with *MAF* and *MAFB* translocations in multiple myeloma, and such translocations are markers of poor prognosis[42,43]. Of three samples with *MAF--IGL*, each had outlier APOBEC signature scores and high *MAF* expression, lending further evidence to the relationship between APOBEC and dysregulated *MAF* (see "Methods").

## Discussion

Our study forms an MM gene fusion landscape and explores clinical relevance. We analyzed the gene expression patterns of fusions, fusions involving kinase genes, druggable targets, evolution of tumor fusion profiles, and translocation and fusion breakpoints of events. We also compared fusions from serial clinic visits and from different tissue sources. We developed methods to map scRNA-seq fusion events to single cells. Our results represent a resource for future studies involving gene fusions in multiple myeloma and other cancer types and highlights several fusion analysis methods. We have built upon prior studies and hope our resource and strategies can be useful for future research and clinical translation.

Targeted sequencing can generate cost-effective reports with clinical utility, including somatic mutations, indels, translocations, and gene expression profiles[44]. Including fusions will require tool development to meet clinical standards, although methodological and study design improvements are being made in this direction[45]. scRNA-seq and long read sequencing will further delineate genomic changes during tumor progression, elucidating subclonal heterogeneity and contextualizing common patterns observed from bulk sequencing.

MM immunotherapies, including checkpoint inhibition, monoclonal antibodies, and chimeric antigen receptor T cells, represent the forefront of targeted therapy. Pan-cancer studies showed reduced mutational load in patients with driver fusions, meaning they would not be ideal candidates for neoantigen-based immunotherapy[8,46]. However, dramatic responses to immunotherapy have sometimes been observed using gene fusions as neoantigens[47].

In multiple myeloma, fusions represent an area for continued study, especially as they relate to gene expression, disease progression, tumor evolution, and targeted therapy. Ongoing research to improve fusion detection tools and pipelines that leverage information from multiple data types will enable more complete pictures of patient tumors as bioinformatics analyses become more deeply integrated into clinical decision making.

## Methods

**Alignment**. Paired RNA-seq fastq files were aligned to GRCh37 using STAR version 2.5.3a_modified[48]. BAM files were sorted and analyzed with flagstat using Samtools version 1.5[49]. Quality control was conducted using FastQC version 0.11.5. (See http://bioinformatics.babraham.ac.uk/projects/fastqc/).

**Association testing and correlation**. Association testing was done using Student's *t* test (two-sided) (continuous expression) and Fisher's Exact Test (two-sided) (categorical expression). Clinical associations with fusions and fusion genes were calculated using Fisher's Exact Test (two-sided) for categorical variables and Mann–Whitney U Test for continuous variables. Expression and clinical testing *p* values were corrected using the Benjamini and Hochberg false discovery rate (FDR) method[50]. All correlations are calculated as Pearson correlations unless otherwise stated.

**Copy number variation detection**. We detected copy number variation from WGS data using BIC-seq2[51] (BICseq2-norm version 0.2.4; BICseq2-seg version 0.7.2). In scRNA-seq, we used inferCNV (version 0.8.2) to calculate single cell copy number profiles[52].

**Fusion analysis scripts**. Fusion results were analyzed by scripts written in Python (version 3.7.2) and R (version 3.5.3). Python packages included numpy, os, and pysam. R packages included ggrepel, gridExtra, readxl, RColorBrewer, Seurat (version 3.0.0), survival, survminer, tidyverse, and UpSetR. (Please see github.com/ding-lab/griffin-fusion/tree/master/mmrf_fusion for fusion analysis scripts.)

**Fusion detection**. We used five fusion detection tools including EricScript[53] (version 0.5.5), FusionCatcher[54] (version 1.00), INTEGRATE[55] (version 0.2.6, using RNA-seq samples only, not paired RNA and WGS), PRADA[56] (version 1.2), and STAR-Fusion[57] (version 1.1.0). Gene names from immunoglobulin super-loci were condensed to IGH, IGK, and IGL (including *IGLL5*).

**Fusion filtering**. Fusions were required to be called by at least two tools. Fusions called by any combination of EricScript, FusionCatcher, or INTEGRATE must also have been called by STAR-Fusion or PRADA in another sample (soft filter tag EFI). Fusions were removed if: partners are the same gene; genes appear on blacklist or are paralogs; fusion comes from list of normal panel fusions (non-cancer cell lines, GTEx, TCGA normal samples)[8,58]; one partner is promiscuous with 25 or more partners (soft filter tag Many Partners); or partner genes are within 300 Kb (soft filter tag within 300 Kb). In addition, across all samples for a particular fusion pair, we required at least one sample to have two or more junction reads or one sample to have one or more spanning reads, or that fusion pair was removed from all samples (soft filter tag Low Count). Finally, fusions with a low WGS support rate compared with the background rate were removed if the binomial test two-sided *p* value was less than 0.15 (soft filter tag Undervalidated). See Supplementary Data 6 for a list of all soft filtered fusions and why they were filtered.

**Gene expression**. Transcripts per million (TPM) was calculated using kallisto[59] (version 0.43.1).

Gene level TPM was calculated as the sum of TPM values from each of that gene's transcripts.

Log transformation of TPM values was calculated as log10(TPM + 1).

**Kinase domain analysis**. Kinase domain status was determined based on reported gene fusion breakpoints using AGFusion[60] (version 1.231). (See http://github.com/murphycj/AGFusion). Following manual review, 15 out of 19 *MAP3K14* fusions were found to possess an intact kinase domain after initially being reported as having disrupted kinase domains due to a lack of annotation.

**Mutation signature profiling**. We used SignatureAnalyzer[61] to quantify mutation signatures.

**Outlier detection**. Gene expression outliers were defined as having values greater than 75th + 1.5*IQR or less than 25th–1.5*IQR, where 75th and 25th represent the 75th and 25th percentile, respectively, and IQR is the interquartile range, defined as the 75th percentile minus the 25th percentile.

**Single cell fusion detection–Fuscia**. Given an aligned BAM file, barcode information for each read mapping to fusion gene regions was extracted using the Python module pysam (version 0.15.2), which wraps Samtools[49] (version 1.7). When two reads map to different genes or regions and share the same cell and molecular barcode, we labeled that transcript as a "chimeric transcript". Multiple reads could originate from the same chimeric transcript. We eliminated reads with length >128 and then selected one representative read from each side of the chimeric transcript by picking the reads mapping closest to the known WGS breakpoint. Transcript overexpression makes false positive detection of chimeric transcripts more likely. We reduced this risk by purposefully looking for chimeric transcripts that may be detected due to overexpression. In plasma cells with IGH translocations, we specifically looked for chimeric transcripts linking IGH and plasma cell markers *SDC1*, *SLAMF7*, and *TNFRSF17*. We called those regions "overlap" regions because chimeric transcripts from genes not associated with fusions overlap with those from legitimate fusions. (Please see http://github.com/ding-lab/fuscia).

We used R (version 3.5.3) and the Seurat[62] package (version 3.0.0) to analyze cell type and gene expression from individual data. Dimensional reduction was performed using UMAP[63].

**Single cell RNA-sequencing data collection**. Additional multiple myeloma patients not related to the MMRF CoMMpass Study were enrolled at Washington University in a longitudinal study. The Washington University Institutional Review Board approved the study protocol, and all relevant ethical regulations, including obtaining informed consent from all participants, were followed.

Five patients (8 samples) from the Washington University IRB-approved study were included in this analysis based on having a translocation relevant to single cell fusion detection (t(4;14), t(8;14), or t(11;14)). Single cell RNA sequencing was conducted using the 10x Genomics Chromium Single Cell 3′ v2 or 5′ Library Kit and Chromium instrument. Approximately 17,500 cells were partitioned into nanoliter droplets to achieve single cell resolution for a maximum of 10,000 individual cells per sample. The resulting cDNA was tagged with a common 16nt cell barcode and 10nt Unique Molecular Identifier during the RT reaction. Full length cDNA from poly-A mRNA transcripts was enzymatically fragmented and size selected to optimize the cDNA amplicon size (approximately 400 bp) for library construction (10x Genomics). The concentration of the 10x single cell library was accurately determined through qPCR (Kapa Biosystems) to produce cluster counts appropriate for the HiSeq 4000 or NovaSeq 6000 platform (Illumina). 26 × 98 bp (3′ v2 libraries) or 2 × 150 bp (5′ libraries) sequence data were generated targeting between 25K and 50K read pairs/cell, which provided digital gene expression profiles for each individual cell. For all the samples included in this study, only Patient 27522 Relapse-2 was processed with the 5′ Library Kit.

**Somatic mutation calling**. MMRF exome bams were aligned to hg19, and somatic variants were called by our in-house pipeline SomaticWrapper, which includes four established bioinformatic tools (Mutect[64] (version 1.1.7), Pindel[65] (version 0.2.54), Strelka2[66] (version 2.9.2), and VarScan2[67] (version 2.3.83)). (See github.com/ding-lab/somaticwrapper.) We kept SNVs called by at least two out of three tools (Mutect, Strelka, VarScan2). Likewise, we kept INDELs called by at least two out of three tools (Pindel, Strelka, VarScan2). We required 14X coverage for somatic mutation calls and only kept mutations with tumor variant allele frequency (VAF) > = 0.05 and normal VAF < = 0.02.

**Structural variant detection**. Structural variants were detected from paired normal and tumor WGS samples using Delly[68] (version 0.7.6) and Manta[69] (version 1.1.0). To be analyzed, tumor and normal WGS samples must have had matching sequencing assays and a corresponding RNA-seq sample.

**Survival analysis**. We performed survival analysis using progression-free survival as the outcome using the survival (version 2.44-1.1) and survminer (version 0.4.6) packages in R. To test for significant improvements in model fit with additional covariates, we implemented a chi-squared test using the anova function and compared the new model to the baseline model. Only patients whose primary sample corresponded to the pre-treatment clinic visit were included for survival modeling.

**Tumor purity**. We used the R package estimate[70] (version 2.0) to quantify tumor purity from RNA-seq data. Tumor purity of peripheral blood (PB) samples was not quantified.

**WGS support of fusion events**. We used WGS data to determine if reported fusions also had genomic support. We defined a breakpoint window centered at each fusion breakpoint. If there were three or more discordant read pairs mapping to within 100 Kb of each breakpoint, we determined the fusion to be supported by WGS. Reads were filtered by Samtools[49] (version 1.5) with flags -F 1920 -f 1 -q 20. We removed fusions from all samples if the fusion-specific support rate differed significantly from the background support rate of all fusions.

**Reporting summary**. Further information on research design is available in the Nature Research Reporting Summary linked to this article.

## Data availability

Data was provided by The Multiple Myeloma Research Foundation (MMRF) CoMMpass (Relating Clinical Outcomes in MM to Personal Assessment of Genetic Profile) Study (NCT01454297). dbGaP Study Accession: phs000748. Data types analyzed in this study were RNA-seq, whole exome sequencing, whole genome sequencing, and clinical information. The MMRF CoMMpass study can be accessed at https://www.ncbi.nlm.nih.gov/projects/gap/cgi-bin/study.cgi?study_id=phs000748.v7.p4. Single cell RNA-seq data used in this study can be accessed at the NCBI under accession code PRJNA627897 [https://www.ncbi.nlm.nih.gov/bioproject/627897].

The source data and scripts underlying all figures are provided at https://doi.org/10.6084/m9.figshare.11941494 (for everything except scRNA data) and https://doi.org/10.6084/m9.figshare.11941506 (for scRNA data).

The remaining data are available in the Article, Supplementary Information, or are available from the author upon reasonable request.

## Code availability

Data analysis scripts and single cell fusion detection methods are available under the MIT license at github.com/ding-lab/griffin-fusion/tree/master/mmrf_fusion and github.com/ding-lab/fuscia.

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

## Acknowledgements

This work has been supported by the Paula C. and Rodger O. Riney Blood Cancer Research Initiative Fund to L.D. and R.V. and NCI U24CA211006 and U2CCA233303 funds to L.D. We also thank the patients, families, and professionals who have contributed to the Multiple Myeloma Research Foundation CoMMpass Study.

## Author contributions

L.D. and R.V. led project design. S.M.F. led data analysis, wrote manuscript, and generated figures. S.M.F, Q.G., and L.Y. performed fusion detection. C.J.Y. acquired sequence data. S.M.F., Q.G., Y.L., H.S., and C.J.Y. generated RNA-seq expression, CNV, somatic mutation, and SV profiles. S.C. and R.G.J. advised on single cell data and methods. M.A.F., J.K., and D.A.K. managed MMRF and in-house sample data collection. L.D., M.A.F., S.M.F., and R.V. reviewed the manuscript.

## Competing interests

The authors declare no competing interests.
