## [Peer Review File · Nature Communications]

Reviewers' comments:

Reviewer #1 (Remarks to the Author): Expertise in MM and transcriptomics including fusions

This cohort is really surprising, since with a median follow up of only 2.2 years, more than 60% of the patients have already progressed. This fact encourages to be very cautious with the interpretation of the clinical correlations.

The very large majority of the data presented in this paper are not novel and were mainly published in Nature Communications papers. All the "findings" on the IGH-WHSC1/t(4;14) patients were already known.

This reviewer does not understand what is the utility of single cell analyses in a few patients.

Reviewer #2 (Remarks to the Author): Expertise in MM -omics

In this paper Foltz and colleagues have highlighted the importance to detect fusion genes in multiple myeloma as predictors of patient survival and targets of specific therapies. The cohort on which the study is based belongs to the CoMMpass dataset where a high number of newly diagnosed multiple myeloma patients have been enrolled, followed-up already for several years and well characterized from a clinical and biological point of view and represents a very good fully annotated cohort to analyze.

The paper aspires to bring a new methodology to more deeply characterize fusion genes in multiple myeloma, to better stratify patients and to drive targeted therapy. Overall, the paper gives some novel hints on myeloma biology, nevertheless there are some major and minor points that need to be addressed.

Major comments:

Figure 2A: Can the authors use WGS data and seqFISH to find out if the 12 WHSC1 over-expressing samples without a fusion detected have at least a t(4;14) translocation on the DNA side? This also relates to what stated in lines 243-244.

Also, what is the evidence of PAX5, RARA, TXNIP to be driver genes in MM?

lines 208-230: these two paragraphs are not clear. Please reformulate.

In particular:

- Lines 213-220: it is not clear how the Cox proportional hazard model was built and how ANOVA test was able to implement this. Moreover, the ANOVA test is not reported in the statistical methods section and is not specified which kind of ANOVA test was performed. Furthermore, the results are not shown. Please make a table showing the results of these different tests.
- "IGH-WHSC1 and PVT1 fusions were significantly associated with poor prognosis": this sentence lacks of any statistical analysis and result, able to corroborate the assumption.
- Please specify which are the cut-off able to identify population with different "total fusion burden"; moreover, the PFS, even if statistically significant, lacks of the written value of the HR; nevertheless, its impact seems to be almost irrelevant being the confidence interval restricted between 1 and 1.04.
- Lines 222-230: the definition of double-hit multiple myeloma given by the authors is referred to the mSMART criteria, the reference 31 is not correct in this sentence. Nonetheless, the cohort to which the double-hit population is compared is not clear and neither the statistical significance of the result. The same is true about the sub-analysis made on the t(4;14).
- "Ongoing research with larger sample sizes and longer follow-up will enable more robust survival modeling utilizing genomic events to define progression and overall survival risk in MM." This is partially true, due to already published studies that have demonstrated the impact of genomic events in the patients survival (Walker et al, JCO, 2015; Bolli et al., Leukemia, 2018; Walker et al., Blood, 2018).

Fig. 4B: The data reported are very interesting but need to be better specified. The authors show that only 4.3% of plasma cells mapped the fusion gene. Are the other plasma cells clonal? Before assuming a low sensitivity of the methodology, it would be worth to investigate how many of the 2477 plasma cells are clonal. Given this "regional gene expression variation" and the concordance between FGFR3 expression and a lower FGFR3 copy number as shown in Fig. 5B, the authors could infer the CNV variation also at a single-cell level.

Fig. 5C: The text does not correspond to the figure. In the text the authors speak about a multivariate Cox model and the figure represents a univariate Kaplan-Meier curve. Moreover, this KM curve does not add any novelty due to the poor prognosis of t(4;14) is already extensively described. Please correct the figure showing the results of the multivariate analysis mentioned. Moreover, please specify which kind of Cox model has been used and which covariates have implemented this model.

Figure 6A: since expression of MYC when mutated or fused still falls within the top half of non fused samples, would it be worth to investigate MYC copy number status as an additional predictor of expression levels to add resolution to this analysis?

Figure 6B: the discordance between MYC and PVT1 fusions in survival is puzzling. To confirm their prognostic relevance, more data should be added to the multivariate analysis, including the cytogenetic category and recurrent aneuploidies of prognostic value. Have IGLL5 mutations been checked in this setting?

Figure 7 is highly descriptive and in the absence of experimental data it should probably go as supplement. The apobec part is the more intriguing one, as there are a number of outlier cases that are not explained by MAF. Can the authors check more carefully on those (number of fusions, other recurrent translocations)? In fact, the prognostic role of APOBEC is associated with MAF but its hyperactivity has independent prognostic value (maura et al, leukemia 2018) and the biological bases of this are quite unclear.

Minor comments:

Chapter 1 (lines 115-130): summarize all clinical characteristics in a table.

Chapter 2 (lines 144-153): these data are mostly already reported and analyzed as from a biological as from clinical point of view by Mikulasova et al., Haematologica 2019. In this paper the authors deeply analyzed MYC and PVT1 translocation giving also insights on gene expression profile of the two translocated genes and other gene over-expressions possible connected with these structural events. Furthermore, Misund et al Leukemia 2019 provide additional biological and clinical insight in this event.

Please include the papers in the citations and reformulate the lines 151-153.

Fig. 2A, line 185: "High fusion gene expression indicates that the fusion may play a role in cancer development." This is rather vague and not amenable to experimental validation. Can the authors at least highlight a statistically significant difference between levels of genes with and without the translocation?

Fig. 3A: The coefficient of correlation is missing.

Fig 3B: The data presented are very interesting and also properly depicted. However, a correlation test between the number of fusions called from BM and PB samples seems not the right test since what we are looking for is the concordance of the two samples in terms of shared fusions.

Reviewer #3 (Remarks to the Author): Expertise in MM sc-seq

The authors address gene fusion events in myeloma, and try to connect this to clinical outcomes. While this is an interesting approach to pursue, the following comments need to be considered before resubmission to a more narrow-scope journal:

1. The title is too general, and does not represent the key elements of the study. Alternatives like: "Druggable IGH fusions in t(4:14) myeloma" or "A deep analysis of the fusion events in t(4:14) myeloma" would be more appropriate.
2. The theme of "tumor evolution" in the paper is very lacking. The descriptive paragraph on four patients with known common passenger mutations that accumulate along tumor progression in relapsed myeloma, is poorly linked to the fusion events. The APOBEC signature is very under-developed.
3. The 3' scRNAseq data from one (n=1) patient does not add anything to the manuscript. It merely demonstrates the weakness of the 3' scRNAseq method to capture gene fusions.
4. The co-occurrence of IGH fusions with FGFR3 mutations and WHSC1+FGFR3 overexpression should be further developed. The manuscript would be so much stronger with some model/mechanism that can be studied in a cell line system.

Reviewers' comments:

Reviewer #1 (Remarks to the Author): Expertise in MM and transcriptomics including fusions

This cohort is really surprising, since with a median follow up of only 2.2 years, more than 60% of the patients have already progressed. This fact encourages to be very cautious with the interpretation of the clinical correlations.

The very large majority of the data presented in this paper are not novel and were mainly published in Nature Communications papers. All the “findings” on the IGH-WHSC1/t(4;14) patients were already known.

This reviewer does not understand what is the utility of single cell analyses in a few patients.

We appreciate the reviewer's comments. We have striven to highlight the novel aspects of our fusion analyses, and we have updated key parts of the manuscript to emphasize this more clearly. Our results represent the largest genome wide fusion analysis of multiple myeloma patients, with the added novelty of fusions detected at serial clinical visits. To our knowledge, we are the first to compare fusion results across multiple clinical visits as well as from bone marrow and peripheral blood samples, and to present changes in the fusion landscape in the context of tumor evolution.

We have now more carefully articulated our results related to progression-free survival to state they are a reflection of this cohort, which may differ from the broader population of myeloma patients. We have further been more explicit in reporting our progression free survival results, including sample sizes, covariates, and numbers of events (new supplementary figure).

Our scRNA fusion detection method is a new approach for finding fusion events at single cell resolution. With this new direction, there are many long-term implications, including better understanding of tumor heterogeneity, subclonal structure, and the relationship of fusion events with gene expression patterns and other somatic alterations. Our approach to scRNA fusion detection utilizes unique cell and molecular barcodes to identify transcripts with reads mapping to multiple genes. Based on the structure of such chimeric transcripts, we can infer the presence of fusion events and map them to the individual cell of origin. Led by reviewer feedback, we expanded our analysis of single cell fusions to include more patients (8 samples from 5 patients). In-house patients were included based on complementary evidence of translocations from WGS data, including one patient with t(4;14), two with t(8;14), and three with t(11;14). The results reflect patterns learned from bulk analysis but with additional, informative detail. In the patient samples with an initiating t(4;14), fusion events are readily detected, and map to specific malignant plasma cell subclones. In the patient with a secondary t(8;14) event, the t(8;14) subclone appears to be lost at relapse, emphasizing patterns of tumor heterogeneity and treatment response. Finally, t(11;14) events are typically not reported at the RNA level, and we find a similar low detection rate in scRNA.

The Fuscia tool scripts are publicly available at <https://github.com/ding-lab/fuscia> and we encourage future open source development.

To build upon the established narrative of t(4;14), IGH--WHSC1 fusions, and FGFR3 mutation co-occurrence, we performed additional analysis and found that FGFR3 mutations likely occur on the same allele as the translocation. Other clinically related results, such as the association between APOBEC signature and MAF fusions, were expanded but also moved to supplement in order to focus more on novel aspects of the main text.

We thank the reviewer for encouraging us to re-emphasize the novelty and applications built in to our analysis.

Reviewer #2 (Remarks to the Author): Expertise in MM -omics

In this paper Foltz and colleagues have highlighted the importance to detect fusion genes in multiple myeloma as predictors of patient survival and targets of specific therapies. The cohort on which the study is based belongs to the CoMMpass dataset where a high number of newly diagnosed multiple myeloma patients have been enrolled, followed-up already for several years and well characterized from a clinical and biological point of view and represents a very good fully annotated cohort to analyze.

The paper aspires to bring a new methodology to more deeply characterize fusion genes in multiple myeloma, to better stratify patients and to drive targeted therapy. Overall, the paper gives some novel hints on myeloma biology, nevertheless there are some major and minor points that need to be addressed.

We are grateful for the reviewer's thorough evaluation of our work. Based on their insightful comments, we have performed additional analyses and made specific improvements to the manuscript, which we describe point-by-point below.

Major comments:

Figure 2A: Can the authors use WGS data and seqFISH to find out if the 12 WHSC1 over-expressing samples without a fusion detected have at least a t(4;14) translocation on the DNA side? This also relates to what stated in lines 243-244.

The reviewer raises an interesting question -- what is the reason for those 12 samples having high WHSC1 expression without a reported fusion? Of the 12 samples with no WHSC1 fusion but WHSC1 outlier overexpression, 10 of them have seqFISH data available. Of those 10, 5 have a WHSC1 translocation reported (4 with IGH, 1 with IGK). Each of the 5 with a reported translocation had expression percentile ≥ 0.87 , while each of the 5 without a reported translocation each had expression percentile < 0.87 . We found no additional explanation in seqFISH data for why the 5 without translocation would have elevated WHSC1 expression. Interestingly, the 5 without translocation were not defined as outliers on the log10 expression scale, possibly meaning that outlier testing on the log10 scale more accurately relate to underlying biological events driving expression. We have included this additional analysis in the text to emphasize the joint role of expression, fusion status, and WGS integration.

Also, what is the evidence of PAX5, RARA, TXNIP to be driver genes in MM?

Thank you for bringing up this point. PAX5, RARA, and TXNIP were included in Figure 2A as examples of pan-cancer driver genes defined by Bailey et al (Cell 2018). We have updated the manuscript to define this more clearly and state that these genes come from pan-cancer study and have not necessarily been defined as driver genes in multiple myeloma.

lines 208-230: these two paragraphs are not clear. Please reformulate.

In particular:

- Lines 213-220: it is not clear how the Cox proportional hazard model was built and how ANOVA test was able to implement this. Moreover, the ANOVA test is not reported in the statistical methods section and is not specified which kind of ANOVA test was performed. Furthermore, the results are not shown. Please make a table showing the results of these different tests.

Thanks to the reviewer for specific questions about model building and testing. We have carefully revised this section to clarify the methods, and we created a new supplemental figure to include the results of our multivariate survival analysis. In summary, for each gene with at least ten fusions reported, we built a multivariate Cox proportional hazards model with age, stage, and fusion status as covariates and compared it against a baseline model with only age and stage as covariates. Our goal was to identify those fusions which provided a significant improvement in the model fit after accounting for the additional fusion covariate. Since our nested models were based on the same data set, we could compare the likelihood of the fusion model against the baseline model using a chi-square test with 1 degree of freedom implemented with the `anova(model1, model2, test = 'Chisq')` function in the R survival package. We have added this description in the Methods section.

- “IGH--WHSC1 and PVT1 fusions were significantly associated with poor prognosis”: this sentence lacks any statistical analysis and result, able to corroborate the assumption.

We have included the results for our survival analysis in the new supplemental survival figure and included the estimation and significance of our model in the main text.

- Please specify which are the cut-off able to identify population with different “total fusion burden”; moreover, the PFS, even if statistically significant, lacks the written value of the HR; nevertheless, its impact seems to be almost irrelevant being the confidence interval restricted between 1 and 1.04.

We agree with the reviewer that, in this case, a statistically significant result with low HR may seem irrelevant. After careful consideration, we have revised the description of the survival analysis of the total fusion burden. Total fusion burden was defined as the number of reported fusions in a given sample, not binarized as high or low at a particular threshold. The take home message we want readers to get is that the detection of each additional fusion is associated with a significant yet modest decrease in progression-free survival. We have included this result in the new supplemental survival figure and included the relevant model information in the text.

- Lines 222-230: the definition of double-hit multiple myeloma given by the authors is referred to the mSMART criteria, the reference 31 is not correct in this sentence. Nonetheless, the cohort

to which the double-hit population is compared is not clear and neither the statistical significance of the result. The same is true about the sub-analysis made on the t(4;14).

We thank the reviewer for the careful reading of this section and pointing out the discrepancy in double hit definitions used in the same sentence. We have stated this difference more explicitly in the text and made clear what definition we have used in our analysis. Given the small sample size associated with each defined subgroup, we chose to emphasize the disparate median times to progression, which we hope will serve as motivation for future studies with larger sample sizes and greater power to reliably discern differences between subgroups.

- “Ongoing research with larger sample sizes and longer follow-up will enable more robust survival modeling utilizing genomic events to define progression and overall survival risk in MM.” This is partially true, due to already published studies that have demonstrated the impact of genomic events in the patients survival (Walker et al, JCO, 2015; Bolli et al., Leukemia, 2018; Walker et al., Blood, 2018).

We appreciate the reviewer pointing out these additional studies. We have incorporated them into the text as examples of how new studies continue to elucidate clinical and genomic aspects of MM.

Fig. 4B: The data reported are very interesting but need to be better specified. The authors show that only 4.3% of plasma cells mapped the fusion gene. Are the other plasma cells clonal? Before assuming a low sensitivity of the methodology, it would be worth to investigate how many of the 2477 plasma cells are clonal. Given this “regional gene expression variation” and the concordance between FGFR3 expression and a lower FGFR3 copy number as shown in Fig. 5B, the authors could infer the CNV variation also at a single-cell level.

We appreciate the reviewer’s interest and concern regarding the sensitivity of our single cell fusion detection approach, i.e. whether our detection rate might be better defined if we specifically look in cells most likely associated with the translocation event. To address this, we sought to identify a subpopulation of tumor plasma cells carrying t(4;14). To avoid potential confounding, we did not rely on measures of WHSC1 and FGFR3 expression to do this, but rather identified cells with deletion of chromosome 13, which is another early event in this patient’s tumor history. This way, we could be confident that our query population comprised of only malignant plasma cells. Setting the chr13 CNV threshold at 0.6 (only examining cells with CNV < 0.6), we found a similar percentage (4.6%, 54/1166 cells) of cells with a chimeric transcript detected. This small difference from the original detection rate fits our understanding of this patient’s tumor having an initiating t(4;14) event and few non-malignant plasma cells detected.

CNV detection using scRNA data was performed using inferCNV. We have updated the text with this additional analysis and included inferCNV in the methods section.

We also included scRNA fusion detection analysis of 7 additional samples (8 total samples) from 5 patients (each with a known t(4;14), t(8;14), or t(11;14) translocation). Those results are now included as supplemental material. Specifically, the results reflect patterns learned from bulk analysis but with additional, informative detail. In the patient samples with an initiating t(4;14), fusion events are readily detected, and map to specific malignant plasma cell subclones. In the patient with a secondary t(8;14) event, the t(8;14) subclone appears to be lost at relapse, emphasizing patterns of tumor heterogeneity and treatment response. Finally, t(11;14) events are typically not reported at the RNA level, and we find a similar low detection rate in scRNA. In the additional samples reported, we included all chimeric transcripts, not excluding chimeric transcripts that mapped to regions with more chimeric transcripts by chance.

Fig. 5C: The text does not correspond to the figure. In the text the authors speak about a multivariate Cox model and the figure represents a univariate Kaplan-Meier curve. Moreover, this KM curve does not add any novelty due to the poor prognosis of t(4;14) is already extensively described. Please correct the figure showing the results of the multivariate analysis mentioned. Moreover, please specify which kind of Cox model has been used and which covariates have implemented this model.

We have updated Figure 5C to be the corresponding multivariate model forest plot rather than the univariate Kaplan-Meier curve. This plot shows each of the covariates included in the model, and we have been more explicit in the text to specify that our model is a multivariate Cox proportional hazards model and also state what variables are included.

Figure 6A: since expression of MYC when mutated or fused still falls within the top half of non fused samples, would it be worth to investigate MYC copy number status as an additional predictor of expression levels to add resolution to this analysis?

We found that MYC copy number was a marginally significant predictor of MYC gene expression in a multivariate linear regression model with MYC/PVT1 fusion status, MYC mutation status, and copy number as predictors of gene expression. Looking deeper, we added MYC copy number to our progression-free survival model but did not observe a significant improvement in the model fit (chi-square test, described previously). We saw the same lack of improvement after including MYC expression in the survival model. Based on this analysis, we decided to favor the more parsimonious original model and not include MYC copy number or expression level in the progression free survival model.

Figure 6B: the discordance between MYC and PVT1 fusions in survival is puzzling. To confirm their prognostic relevance, more data should be added to the multivariate analysis, including the cytogenetic category and recurrent aneuploidies of prognostic value. Have IGLL5 mutations been checked in this setting?

We tested for the overlap between MYC--IGL or PVT1--IGL fusion and IGLL5 mutation but only found two such patients, so we did not include IGLL5 as a covariate for further

testing. However, we did investigate high/intermediate risk cytogenetic categories of prognostic significance, namely del(17p), amp(1q), t(4;14), t(8;22), t(14;16), and t(14;20), as well as standard risk Hyperdiploid status to discern any additional insights that could help explain the observed patterns. See table below for patients with complete data.

Fusion	Mutant	Total	del17p	amp1q	t(4;14)	t(8;22)	t(14;16)	t(14;20)	Hyper-diploid
None	None	435	56	143	52	19	14	7	253
None	MYC	8	0	7	1	0	4	0	3
MYC--IGL	None	8	1	3	2	1	0	0	6
PVT1--IGL	None	7	1	4	1	6	0	0	6

Of the 15 patients with a MYC--IGL or PVT1--IGL fusion and complete seqFISH data for this comparison, 8 had MYC--IGL and 7 had PVT1--IGL. From the 8 with MYC--IGL, only 1 had t(8;22) (translocation between MYC and IGL) according to seqFISH. Of the 7 with PVT1--IGL, 6 had t(8;22) according to seqFISH. This discrepancy is further reflected in the WGS support summary in Fig. 1b. We interpret this result to mean that fusions annotated as PVT1--IGL may be more closely associated with translocations annotated as t(IGL,MYC). Barwick et al. (2019 Nature Communications) recently showed the prognostic significance of t(8;22), especially as a secondary event in myeloma development. However, this does not explain the observed protective trend seen with reported MYC--IGL fusions in the absence of detected t(8;22). Incidentally, the one MYC--IGL patient with reported t(8;22) was diagnosed with Stage III disease at age 50 and had progression-free survival of 245 days, which fits the survival timeline of the other t(8;22) patients more than the non-t(8;22) patients. To the extent that PVT1--IGL detection acts as a proxy for t(8;22), this fusion has prognostic value. We hesitate to make further conclusions about the MYC--IGL fusion group due to the censoring pattern and small sample size of those patients. Future studies specifically targeting this phenomena are necessary. We have updated the manuscript text to delve into this topic more. Thank you for encouraging us to investigate this ongoing discussion more deeply.

Figure 7 is highly descriptive and in the absence of experimental data it should probably go as supplement. The apobec part is the more intriguing one, as there are a number of outlier cases that are not explained by MAF. Can the authors check more carefully on those (number of fusions, other recurrent translocations)? In fact, the prognostic role of APOBEC is associated

with MAF but its hyperactivity has independent prognostic value (maura et al, leukemia 2018) and the biological bases of this are quite unclear.

We appreciate the reviewer's suggestion regarding the placement of Figure 7. We have taken the reviewer's advice and moved it to supplement.

We expanded upon our previous APOBEC findings. The correlation between APOBEC score and total number of fusions is 0.0155, so there does not appear to be any relationship between APOBEC score and fusion events like there is for mutation events. Out of 40 patients with outlier APOBEC score, 35 have seqFISH data. 23/35 have an IG-MAF or IG-MAFB translocation and 12 do not. Of the 12 without MAF/B translocation, 1 has MAF--IGL fusion and one has YBX1--PPA1 fusion (see below). In addition to 3 MAF--IGL samples which all have outlier APOBEC scores, samples with fusions in two other genes have a median APOBEC score above the outlier threshold of 0.302: C22orf34 and YBX1. We have included these additional genes in the (now supplemental) figure as well as some discussion about what is known regarding the connection between YBX1 and APOBEC.

Minor comments:

Chapter 1 (lines 115-130): summarize all clinical characteristics in a table.

We have made textual references to the supplementary table presenting clinical characteristics more explicit.

Chapter 2 (lines 144-153): these data are mostly already reported and analyzed as from a biological as from clinical point of view by Mikulasova et al., Haematologica 2019. In this paper the authors deeply analyzed MYC and PVT1 translocation giving also insights on gene expression profile of the two translocated genes and other gene over-expressions possible connected with these structural events. Furthermore, Misund et al Leukemia 2019 provide additional biological and clinical insight in this event.

Please include the papers in the citations and reformulate the lines 151-153.

We appreciate the reviewer pointing us to these additional studies. We have duly incorporated them into the text.

Fig. 2A, line 185: "High fusion gene expression indicates that the fusion may play a role in cancer development." This is rather vague and not amenable to experimental validation. Can the authors at least highlight a statistically significant difference between levels of genes with and without the translocation?

Following the reviewer's suggestion, we have rearranged statements concerning overexpression of fusion genes and removed vague language.

Fig. 3A: The coefficient of correlation is missing.

We have added the correlation coefficient to the bottom right corner of the panel.

Fig 3B: The data presented are very interesting and also properly depicted. However, a correlation test between the number of fusions called from BM and PB samples seems not the right test since what we are looking for is the concordance of the two samples in terms of shared fusions.

Thank you for your attention to both the testing and visualization methods. We have implemented an additional metric (normalized Hamming distance) to characterize the overlap between the BM and PB fusion calls and included this as part of Fig. 3b. We agree that the correlation between the number of calls from BM and PB does not intuitively convey the overlap between the samples, and we have now removed this comparison from the text.

Reviewer #3 (Remarks to the Author): Expertise in MM sc-seq

The authors address gene fusion events in myeloma, and try to connect this to clinical outcomes.

While this is an interesting approach to pursue, the following comments need to be considered before resubmission to a more narrow-scope journal:

We appreciate the reviewer's comments and willingness to provide feedback on our manuscript. We have included our responses to each suggestion below.

1. The title is too general, and does not represent the key elements of the study. Alternatives like: "Druggable IGH fusions in t(4:14) myeloma" or "A deep analysis of the fusion events in t(4:14) myeloma" would be more appropriate.

Thank you for your suggestions of alternative titles -- after careful consideration we have updated the title to reflect our focus on the structure of fusions related to t(4;14) and t(8;22) as well as structures derived from our single cell analysis. The updated title is "Evolution and structure of clinical relevant gene fusions in multiple myeloma."

2. The theme of "tumor evolution" in the paper is very lacking. The descriptive paragraph on four patients with known common passenger mutations that accumulate along tumor progression in relapsed myeloma, is poorly linked to the fusion events. The APOBEC signature is very under-developed.

We appreciate the reviewer's concern with respect to tumor evolution. Our intention in focusing on how the fusion landscape changes over time within the four patients highlighted in Figure 3c-d was to present possible interpretations of the data while respecting the uncertainties inherent to this data type. We have updated this section of the manuscript to expand upon the potential relationships between mutations and fusion events in these patients.

Thank you for encouraging us to more fully develop the APOBEC signature analysis. We performed additional analyses which are now included in the main text. The main additional finding is that, in addition to 3 MAF--IGL samples, we found that samples with fusions involving C22orf34 and YBX1 had median APOBEC score above the outlier threshold. In response to another reviewer's suggestion, Figure 7 detailing druggable events and APOBEC has been made a supplemental figure.

3. The 3' scRNAseq data from one (n=1) patient does not add anything to the manuscript. It merely demonstrates the weakness of the 3' scRNAseq method to capture gene fusions.

Thank you for raising this concern. After initially reporting one fusion example from one patient, we have now included results of 8 samples from 5 patients, including patients with t(4;14), t(8;14), and t(11;14) translocations. Those findings are now covered in the

manuscript and a new supplementary figure. The results reflect patterns learned from bulk analysis but with additional, informative detail. In the patient samples with an initiating t(4;14), fusion events are readily detected, and map to specific malignant plasma cell subclones. In the patient with a secondary t(8;14) event, the t(8;14) subclone appears to be lost at relapse, emphasizing patterns of tumor heterogeneity and treatment response. Finally, t(11;14) events are typically not reported at the RNA level, and we find a similar low detection rate in scRNA.

Our purpose in reporting our results in scRNA fusion detection was to explore a novel technique and advance future avenues of algorithm development. By highlighting our early findings in this direction, we hope to encourage the community's efforts in an open and collaborative way. Our scRNA fusion detection method is a new approach for finding fusion events at single cell resolution. Long term implications include improving understanding of tumor heterogeneity, subclonal structure, and the relationship of fusion events with gene expression patterns and other somatic alterations. By utilizing unique cell and molecular barcodes and identifying transcripts with reads mapping to more than one gene, we can infer the presence of fusion events and map them back to the original cell. We expanded our analysis single cell analysis to now include more patients (8 samples from 5 patients), each with supporting WGS evidence of translocations. One patient had t(4;14), two had t(8;14), and three had t(11;14). The Fuscia tool scripts are publicly available at <https://github.com/ding-lab/fuscia> and we encourage future open source development.

4. The co-occurrence of IGH fusions with FGFR3 mutations and WHSC1+FGFR3 overexpression should be further developed. The manuscript would be so much stronger with some model/mechanism that can be studied in a cell line system.

We appreciate the reviewer's interest in IGH/WHSC1 fusion and FGFR3 expression/mutation co-occurrence. After considering various ways to explore the co-occurrence of IGH fusions and FGFR3 pathogenic mutations, we compared the variant allele frequencies of FGFR3 mutations in DNA and RNA to understand the relationship between mutation and expression. Our hypothesis was that by comparing the DNA and RNA VAF, we could gain insight into the possible allelic/haplotypic relationship of the mutation and translocation. What we found was a consistent trend of the RNA VAF being two or more times higher than the DNA VAF, each with neutral FGFR3 copy number. Given the allele specific overexpression of the mutant allele, our interpretation is that transcription of the FGFR3 mutant allele is being driven by the 3' enhancer element of IGH, meaning the mutations each occurred in cis with the translocation (on the same der(14) homolog as the 3' IGH enhancer). Future studies with linked-read or long read WGS could potentially confirm this observation. This DNA RNA VAF comparison is now included as a supplementary figure and the manuscript text has been updated to include this additional result.

Reviewers' comments:

Reviewer #1 (Remarks to the Author):

(Added by Ed) This reviewer did not provide any comments for the authors

.

Reviewer #2 (Remarks to the Author):

In the revised version of the paper, the authors have improved most aspects of their work. However, few unsolved issues remain.

Line 213: we appreciate a better detail on the multivariate analysis. However, many more prognostic factors should be tested against fusions: among those, R-ISS and major cytogenetic abnormalities.

Lines 224-226: can the authors visualize the additive effect of each gene fusion? For instance, through a forest plot or a Kaplan Meier plot

Lines 231-232: the authors again refer to the "double hit" in a wrong fashion: according to Walker et al, double hit events are TP53 bi-allelic inactivation or ISS III AND chr(1q)amp.

Reviewer #3 (Remarks to the Author):

The authors have improved their analyses, and the manuscript is better communicated to the reader.

Still needs improvement:

1. 3' scRNAseq data still shows very low sensitivity for gene fusions. Increasing the n won't help. This should move to the supp data, as it does not support the main findings of the paper. Alternatively, the authors can perform a full length scRNAseq protocol.
2. I still do not understand what is the relationship to the APOBEC signature, and the relevance to the main findings. I think this piece should be removed.
3. The reviewers remark on 11:14 fusion event that is not detected by RNA is inaccurate at best. In 90% of cases, t(11:14) is characterized by CCND1 overexpression.

I have no further comments.

Reviewer #2 (Remarks to the Author):

In the revised version of the paper, the authors have improved most aspects of their work. However, few unsolved issues remain.

We appreciate your comments. Thank you!

Line 213: we appreciate a better detail on the multivariate analysis. However, many more prognostic factors should be tested against fusions: among those, R-ISS and major cytogenetic abnormalities.

Thank you for this suggestion regarding including R-ISS and major translocations as covariates in our multivariate survival models. Following the reviewer's suggestion, we built new models including R-ISS and major translocations as covariates using the following approach.

We calculated R-ISS following these guidelines:

R-ISS I = ISS Stage I, no high-risk cytogenetic events, and normal LDH

R-ISS II = not R-ISS I or R-ISS III

R-ISS III = ISS Stage III, and either high-risk cytogenetic events or high LDH

(High-risk cytogenetic events included TP53 loss, 1q gain, MAF/MAFB translocation, and t(4;14).)

We also included model covariates to control for hyperdiploid status, t(11;14), t(4;14), t(14;16), t(6;14); t(14;20), and IGH/IGK/IGL translocations with MYC.

After including R-ISS and major cytogenetic events, we found no fusions to be significantly associated with PFS. This is likely due to using a smaller subset of patients for testing and the introduction of confounding factors to the model. In our model design, we considered that most recurrent fusions are closely associated with translocation events, and that using both R-ISS and translocations as covariates could potentially introduce confounding to the model. Also, not all patients have WGS data available for determining R-ISS and translocation status, so only a subset of patients (611/691) could be included in the R-ISS/translocation models.

We have included this additional analysis in the revised manuscript.

Lines 224-226: can the authors visualize the additive effect of each gene fusion? For instance, through a forest plot or a Kaplan Meier plot

We edited the manuscript to explicitly refer to forest plot for total fusions in Supplementary Fig. 2d. Thank you for alerting us to this oversight in the manuscript.

Lines 231-232: the authors again refer to the “double hit” in a wrong fashion: according to Walker et al, double hit events are TP53 bi-allelic inactivation or ISS III AND chr(1q)amp.

Thank you -- we have updated the manuscript to accurately describe Walker et al's double hit group having biallelic TP53 inactivation or having Stage III disease with CKS1B amplification.

Reviewer #3 (Remarks to the Author):

The authors have improved their analyses, and the manuscript is better communicated to the reader.

Thank you -- we appreciate your encouragement.

Still needs improvement:

1. 3' scRNAseq data still shows very low sensitivity for gene fusions. Increasing the n won't help. This should move to the supp data, as it does not support the main findings of the paper. Alternatively, the authors can perform a full length scRNAseq protocol.

We appreciate the reviewer's point regarding sensitivity of the 3' scRNA-seq data, and we anticipate the next stage of technology development with full-length scRNA profiling. After careful consideration, we feel it is helpful to retain this section in the main manuscript as we believe that single cell fusion detection methods will be a powerful technique, especially with improved data, and we now acknowledge in the manuscript that the current approach represents an early stage in that process. We have contextualized our findings by including the future direction of using full-length scRNA-seq protocol.

2. I still do not understand what is the relationship to the APOBEC signature, and the relevance to the main findings. I think this piece should be removed.

Thank you for voicing this concern. We have followed the suggestion to remove the APOBEC signature from Supplementary Fig. 8d. In the manuscript, we reworded this section to emphasize how our MAF--IGL fusion finding re-enforces a known relationship between MAF dysregulation and the APOBEC signature.

3. The reviewers remark on 11:14 fusion event that is not detected by RNA is inaccurate at best. In 90% of cases, t(11;14) is characterized by CCND1 overexpression.

We appreciate you pointing out this confusion due to poor wording on our part. We have updated the manuscript to clarify our intended meaning that any potential fusion products associated with t(11;14), if they exist, may not be reported by fusion calling algorithms, despite clear evidence of t(11;14) due to CCND1 overexpression.

I have no further comments.